# Comprehensive Analysis of Age- and Sex-Related Expression of the Chaperone Protein Sigma-1R in the Mouse Brain

**DOI:** 10.3390/brainsci14090881

**Published:** 2024-08-30

**Authors:** Khadija Tarmoun, Véronik Lachance, Victoria Le Corvec, Sara-Maude Bélanger, Guillaume Beaucaire, Saïd Kourrich

**Affiliations:** 1Department of Biological Sciences, Faculty of Sciences, University of Quebec at Montreal, 141 President-Kennedy Street, Montreal, QC H2X 1Y4, Canada; 2Center of Excellence for Research on Orphan Diseases, Courtois Foundation (CERMO-FC), Montreal, QC H2X 3Y7, Canada; 3Center for Studies in Behavioral Neurobiology, Concordia University, Montreal, QC H4B 1R6, Canada

**Keywords:** aging, Sigma-1R, chaperone protein, sex, brain, mouse, immunohistochemistry

## Abstract

Sigma-1R (S1R) is a ubiquitously distributed protein highly expressed in the brain and liver. It acts as a ligand-inducible chaperone protein localized at the endoplasmic reticulum. S1R participates in several signaling pathways that oversee diverse cellular and neurological functions, such as calcium and proteome homeostasis, neuronal activity, memory, and emotional regulation. Despite its crucial functions, S1R expression profile in the brain with respect to age and sex remains elusive. To shed light on this matter, we assessed S1R distribution in the mouse brain across different developmental stages, including juvenile, early adult, and middle-aged mice. Using immunohistochemistry, we found that S1R is predominantly expressed in the hippocampus in juvenile mice, particularly in CA1 and CA3 regions. Notably, S1R is not expressed in the subgranular layer of the dentate gyrus of juvenile mice. We observed dynamic changes in S1R levels during development, with most brain regions showing either an abrupt or gradual decline as mice transition from juveniles to adults. Sexual dimorphism is observed before puberty in the hippocampus and hypothalamus and during adulthood in the hippocampus and cortex.

## 1. Introduction

S1R is broadly expressed in brain regions responsible for cognition, mood regulation, and learning, such as the hippocampus, amygdala, cortex, striatum, thalamus, and hypothalamus [1]. As a neuroprotective protein and an inter-organelle signaling modulator, S1R plays a critical role in calcium homeostasis, regulation of neuronal excitability and plasticity, lipid metabolism, autophagy, and protein folding [2,3,4]. Therefore, disruption of S1R functions can take part in countless brain disorders, including but not limited to Alzheimer’s disease (AD), amyotrophic lateral sclerosis (ALS), and neuropsychiatric disorders such as addiction, depression, and anxiety [5,6,7,8,9]. Biological activity of S1R can be modulated by diverse ligands such as psychostimulant drugs (e.g., cocaine and methamphetamine), antipsychotic drugs (haloperidol), and sex steroid hormones (e.g., pregnenolone, dehydroepiandrosterone (DHEA), and progesterone) [10,11,12,13,14,15,16]. As a single transmembrane chaperone residing at the endoplasmic reticulum (ER), S1R forms an inactive complex at the mitochondria-associated endoplasmic reticulum membranes (MAMs) with the chaperone BiP. Agonist-induced activity of S1R or ER stressors prompts the dissociation of the S1R-BiP complex [17], which promotes the translocation of S1R to different compartmental interfaces, such as the ER-nuclear membrane or ER-lipid droplet, where it can regulate a variety of cellular functions [18].

As documented, aging induces progressive structural and functional changes in the brain that disturb cognitive functions [19]. In fact, aging is a well-known and predominant risk factor for many neurodegenerative diseases, such as AD and Parkinson’s diseases (PD) [20]. Interestingly, these conditions exhibit sex-specific disparities in their incidence [21,22,23], and S1R expression has been shown to change between males and females of an AD mouse model [24] Indeed, expression of S1R in the brain has been studied in humans and other animal models, including rodents and primates. On the one hand, positron emission tomography (PET) scan studies conducted on male monkey brains have demonstrated that the radioligand [^11^C]SA4503 binding to S1R increases with age, suggesting that S1R’s binding pocket availability is enhanced during aging [25]. On the other hand, a decrease in S1R binding has been noted in a few areas of male rat brains through PET scanning studies also performed with the tracer mentioned above [1]. However, diaminobenzidine immunohistochemistry (DAB-IHC) experiments executed at different ontogenic stages indicated that S1R expression remains stable in the brain of C57BL/6 male mice [26]. Considering the limited research on sex-specific variability in S1R expression and the inconclusive findings regarding age-related changes in S1R levels in the brain, our study seeks to determine the age- and sex-specific effects on S1R expression in the mouse brain.

## 2. Materials and Methods

### 2.1. Animals

The C57BL/6J mice were sourced from the Jackson Laboratory in Bar Harbor, Maine, and were housed at the animal facility of the Université de Québec à Montréal. The animals were kept under constant conditions, with a temperature of 20–24 °C, humidity levels of 40–60%, and a 12-h light/dark cycle (light on at 7:00 A.M.). They were provided with *ad libitum* access to food and water. Mice were weaned at 19–21 days. Unless mentioned, mice used in this study were from three different cohorts. The brains of 9 male and 9 randomly estrous cycling female mice were used to perform Western blotting experiments. The brains of 9 male and 9 randomly estrous cycling female mice were also used to perform immunohistochemistry experiments. All mice were handled in a facility approved by the Canadian Council on Animal Care in accordance with their Guide for Care and Use of Experimental Animals, ethical approval code 0324-C1R1-964-0325, accepted on 26 March 2023.

### 2.2. Antibodies and Reagents

Ketamine (Narketan^®^, 100 mg/mL, cat# DIN02374994) and xylazine (Rompun^®^, 100 mg/mL, cat# DIN02169606) were from CDMV Inc., Saint-Hyacinthe, QC, Canada. Glycerol (cat# G7893), ethylene glycol (cat# 324558), paraformaldehyde (PFA, cat# 441244), sucrose (cat# S1888), goat serum (cat# S26), Triton-X-100 (cat# T8787), and Ponceau S Solution (#P7170) were from Sigma-Aldrich, St. Louis, MO, USA. Hoechst 33342 dye (cat# H1399), goat anti-rabbit IgG conjugated to Alexa Fluor 647 (cat# Q48285), ProLong Glass antifade mounting reagent (cat# P36980), Pierce™ Protease Inhibitor Mini Tablets, EDTA-free (cat# A32955), goat anti-rabbit IgG F(ab’)2 secondary antibody, HRP (cat# 31461, 1:10 000), Micro BCA™ Protein Assay Kit (cat# 23235), and Lipofectamine 2000 (cat#11668019) were from ThermoFisher Scientific, Waltham, MA, USA. Rabbit anti-SIGMAR1 antibodies (cat# 74807S and cat# 61994S) were from Cell Signaling Technology (Danvers, MA, USA). Western Lightning Plus ECL (#NEL105001EA) was from PerkinElmer, Waltham, MA, USA. Immobilon^®^-FL Transfer membrane 0.45 um and Polyvinylidene Difluoride (PVDF) membrane (#ISEQ00010) were from Merck Millipore, Burlington, MA, USA. OCT compound (#23–730-571) and microscope slides (# 12-550-15) were from Fisher Scientific. The liquid blocker pap pen (#71310) was from Electron Microscopy Sciences (EMS), Hatfield, PA, USA.

### 2.3. Plasmid Constructs

The SIGMAR1 (NCBI accession NM_005866.4) cDNA was amplified by PCR using the following primers: For *5*′-ggc ggt acc atg cag tgg gcc gtg ggc cg-*3*′ and Rev *5*′-ggc ctc gag cg agg gtc ctg gcc aaa gag g. The PCR fragment was digested with KpnI and XhoI and ligated into the pcDNA3.1 V5-6xHis-TOPO TA vector to produce the S1R-V5 plasmid. Site-directed mutagenesis were carried out by PCR using the Q5^®^ Site-Directed Mutagenesis Kit (New England Biolab, cat#E0554S) to introduce silent mutations in the S1R-V5 plasmid and generate a siRNA resistant plasmid (referred to as S1R-V5 r1) by using the following primers: For *5*′-ctc acc ctc ttt tac acc ctc cga tcg tac gct cgg ggc ctc cgg ctt gag-*3*′ and Rev *5*′-gaa gtc ctg ggt gct gaa gac agt gtc ggc-*3*′. Integrity of the coding sequence of these constructs was confirmed by dideoxy sequencing.

### 2.4. Cell Culture and siRNA Assay

Human embryonic kidney HEK293 cells were maintained in DMEM (Dulbecco’s Modified Eagle’s Medium) (Wisent #319-005-CS) supplemented with 10% (*v*/*v*) FBS (fetal bovine serum, Wisent #080-150) at 37 °C in a humidified atmosphere containing 5% CO_2_. The synthetic oligonucleotide Silencer™ Select Pre-Designed siRNA, SIGMAR1 s20088, targeting the *SIGMAR1* gene, and the negative control siRNA (Silencer™ Select Negative Control No. 1, cat# 4390843) were purchased from ThermoFisher Scientific. HEK293 cells were transfected with 20 nm oligonucleotide using the Lipofectamine 2000 transfection reagent according to the manufacturer’s indications except for the following modifications. Cells were seeded directly into the transfection mix at twice the cell density as indicated in the basic protocol. The next day, transient transfection of HEK293 cells grown to 50–70% confluence was performed with the pcDNA3 or S1R-V5 r1 plasmids while using the TransIT-LT1 Reagent (Mirus, #MIR2305) according to the manufacturer’s instructions. Empty pcDNA3 vector was added to keep the total amount of DNA per plate constant. Protein expression analysis by Western blotting was performed at 72 h post-seeding. Briefly, the cells were washed twice with ice-cold PBS and harvested in 200 μL of lysis buffer (150 mm NaCl, 50 mm Tris (pH 8.0), 0.5% deoxycholate, 0.1% SDS, 10 mm Na_4_P_2_O_7_, 1% IGEPAL, and 5 mm EDTA) supplemented with protease inhibitors. After a 60-min incubation in lysis buffer at 4 °C, the lysates were centrifuged for 20 min at 14,000× *g* at 4 °C. The pellet was discarded, and the protein concentration was measured using BCA assay. The extracts were denatured by addition of 35 μL of SB4X (200 mM Tris-HCl pH 6.8, 8% SDS, 4% β-mercaptoethanol, 40% glycerol, 0.4% Bromophenol Blue), followed by a 60-min incubation at room temperature.

### 2.5. Tissue Homogenization

Once the tissue was harvested and flash frozen, 400 μL of homogenization buffer (0.32 M sucrose, 1 mM NaHCO_3_, 20 mM HEPES, 0.25 mM CaCl_2_, 1 mM MgCl_2_ and protease/phosphatase inhibitors) were added to 0.1 g of tissue and homogenized with blue pestle and cordless pestle motor in a 1.5 mL Eppendorf tube. Using an insulin syringe, 20 up and down movements were performed to disrupt the tissue. 10X detergent solution (5% sodium-deoxycholate, 1% SDS, and 10% IGEPAL) was added and diluted to 1X to the samples, and the homogenates were incubated for 30 min. at 4 °C using end over end mixing. Next, the samples were centrifuged at 15,000× *g* for 15 min. at 4 °C, the supernatant harvested, and the pellet discarded. Samples were diluted 1:10, and protein concentration was measured using BCA assay. All biological samples were analyzed in triplicate in three independent experiments. Samples were diluted to 1 mg/mL with homogenization buffer and SB4X denaturation buffer (200 mM Tris-HCl pH 6.8, 8% SDS, 4% β-mercaptoethanol, 40% glycerol, 0.4% Bromophenol Blue) diluted to 1X. Samples were denatured at 95 °C for 5 min and spun at room temperature (RT) for 30 s at 14,000 rpm.

### 2.6. Western Blotting (WB)

From 20–30 μg of the protein samples was separated on 10% SDS-PAGE gels for 60 min. at 150 V at RT using 1X SDS Running buffer (25 mM Tris-Base, 192 mM Glycine and 0.1%SDS, pH 8.3–8.8). The proteins were transferred to a PVDF membrane for 1 h at 100 V at 4 °C. The membranes were stained for 10 min. with Ponceau S. Excess stain was removed for 1 min. with Milli-Q water. The membranes were scanned and blocked in TBS 1X containing 0.1% Tween 20 (TBST) and 5% non-fat dry milk for 1 h at RT. Primary antibodies were applied, and membranes were incubated overnight at 4 °C. The membranes were washed 3 × 8 min. with TBST. Secondary antibodies were applied, and membranes were incubated for 1 h at RT. The membranes were washed 3 × 8 min. with TBST and twice with TBS 1X. The proteins were visualized using ECL detection kit and Fusion-FX imaging system.

### 2.7. Densitometry Analysis

Western blot quantification was performed based on the recommendations of Gassmann et al. [27]. All quantified immunoblots were carefully exposed to avoid saturation. Acquisitions were saved in 16-bits grayscale images. Blots were analyzed using ImageJ software. Lanes were selected and plotted using the ‘Gel analyzer’ functions. Peaks on the plots were individually closed to the background level of each lane using the Straight line’ tool and the enclosed area was measured using the ‘Wand’ tool.

### 2.8. Perfusion and Cryo-Sectioning

The mice were anesthetized with a mixture of ketamine and xylazine, with a dose of 200 mg/kg of ketamine and 20 mg/kg of xylazine, administered via intraperitoneal injection. Following anesthesia, transcardial perfusion was performed using 25–30 mL of ice-cold PBS 1X at a rate of 6–7 mL/min to remove excess blood, followed by perfusion with 25–30 mL of ice-cold 4% PFA. The brains were then removed and further fixed using a 15 mL tube filled with 4% PFA and kept at 4 °C overnight for 16–18 h. The next morning, the brains were washed 3× with 15 mL of PBS 1X and transferred to a new 15 mL tube filled with 30% sucrose and incubated 48 h at 4 °C. Following an additional 3 washes with 15 mL of PBS 1X, the brains were sealed in a 3 MIL plastic and immersed in dry ice powder and frozen for 30 s. Brains were stored at −80 °C for a few days before sectioning. Prior to sectioning, the brain was mounted in OCT compound in the cryostat chamber. Cryo-sectioning was performed using Leica CM1950. Coronal sections of 30 µm were cut and preserved in a cryoprotectant solution (25% glycerol, 30% ethylene glycol, 50 mM phosphate buffer, pH7.2). Coronal sections were mounted on microscope slides. Each slide contained males and females as well as the three age groups. Three technical replicates of each subregion (S1 to S6) of all biological samples were made and analyzed.

### 2.9. Immunohistochemistry (IHC) & Antibody Validation

Detection of S1R in brain sections was carried out using indirect immunofluorescence. First, a liquid blocker pap pen was used to circle the brain sections, which were subsequently permeabilized with 150 μL/section of PBS 1X containing 0.3% Triton-X-100 for 1 h at room temperature. Next, the slices were blocked in PBS 1X + 10% goat serum for 1 h at room temperature. Slices were then incubated in rabbit anti-S1R primary antibody (1/1000) at 4 °C for 24 h. Subsequently, the slides were washed 3 times with the blocking solution, and goat anti-rabbit IgG conjugated to an Alexa Fluor 647 (1/200) was applied and incubated for 1h at room temperature. After 3 washes with PBS 1X, the nuclei were stained with Hoechst 33342 dye (2 μg/mL) for 10 min at room temperature and were protected from light. After another 3 washes with PBS 1X, sections were mounted with ProLong Glass antifade reagent. The specificity of the anti-S1R antibody was confirmed by using an adsorption control. Prior to incubation with the tissue, the S1R primary antibody was blocked overnight at 4 °C with a S1R recombinant C-terminal fragment protein purified in our laboratory. Then, the sections were incubated with the S1R antibody that had its epitopes blocked by recombinant protein.

### 2.10. Confocal Imaging and Image Analyses

Sections were examined using air and oil immersion objectives (20X and 60X) and a Nikon A1R confocal microscope. Image processing and quantification were performed with FIJI software (ImageJ v.1.51r). Regions of interest were manually delineated according to The Mouse Brain in Stereotaxic Coordinates, Second Edition [28]. The signal intensity of each region of interest underwent correction based on the covered area and subsequently was normalized to juvenile male mice. This normalization process aimed to enable straightforward comparison of structures across different age groups and sex.

### 2.11. Statistical Analysis

GraphPad Prism software version 10.2 for Windows was used to conduct statistical analysis. Experimental data were compared using a two-way ANOVA, followed by a post hoc Bonferroni multiple comparison test. Data were considered significant when *p* values were <0.05 (*), <0.01 (**) or <0.001 (***).

## 3. Results

### 3.1. S1R Is Differentially Expressed in Males and Females and Reduces during Aging in the Hippocampus, Cortex, and Striatum

S1R plays a pivotal role in neuronal plasticity, learning, memory, and cognitive decline. To comprehensively explore its impact across critical developmental stages, we focused on specific ontogenic stages ranging from pubertal onset (juvenile, 4 weeks) to early adulthood marked by heightened brain plasticity (8–16 weeks), and advanced adulthood where initial indications of age-related memory decline manifest in mice (middle-aged, 32–40 weeks) [29,30]. To this end, our study involved three cohorts of wild-type C57BL/6J mice, comprising both male and female subjects (n = 18). Following brain extraction and dissection of the hippocampus, isocortex, and striatum, we first sought to investigate S1R expression using Western blot experiments (Figure 1). Interestingly, we noticed a significant age-dependent reduction of S1R expression in male and female hippocampal and cortical tissue. Worthy of note is the sexual dimorphism observed in the juvenile and middle-aged hippocampal and adult cortical tissue. In fact, for all brain regions, female mice showed much less S1R than the age-matched males (Figure 1A–D). Although not significant, we detected an age-related decline of S1R expression in male striatal tissue, whereas a significant decrease was revealed in female samples (Figure 1E,F). Altogether, these data support an age- and sex-dependent decrease of S1R expression during aging.

### 3.2. Histological Analysis of S1R in Brain Tissue and Antibody Validation

Based on the Western blotting data, we further evaluated S1R expression using immunohistochemistry (IHC) experiments to identify and define more precisely the specific brain areas showing a reduced S1R expression during aging. Before doing so, we confirmed the specificity of the S1R antibodies by using siRNA assay and Western blotting experiments. As observed, both antibodies, clone D4 and clone D7, can detect the endogenous protein in cells transfected with the negative control siRNA (siCtrl) (Figure 2A). As expected, transfection of S1R siRNA (siS1R) reduced the expression of the endogenous protein when co-expressed with pcDNA3 plasmid (Figure 2A). Co-expression of the siRNA resistant plasmid, S1R-V5 r1, in S1R-depleted cells further confirms the ability of our antibodies to recognize the S1R protein (Figure 2A). Next, we proceeded with the cryo-sectioning once perfusion and tissue embedding were completed for all three cohorts. Briefly, each brain was sectioned into different regions (S1 to S6, Figure 2B), and the slices were kept in a 6-well plate filled with anti-freeze media. Following sectioning, brain slices were mounted on positively charged microscope slides. Each section (S1 to S6) was performed in triplicate, and the slides were mounted with tissue harvested from male or female brains of each age group (step 4, Figure 2C). Prior to performing the immunolabeling, we tested the specificity of the primary and secondary antibodies. To do so, we first used a blocking peptide to block the antigen recognition site of the S1R primary antibody. As observed, we were able to block most of the signal related to S1R labeling when comparing tissue incubated with primary and secondary antibodies (Figure 2D) to the tissue exposed to the primary antibody blocked by the peptide and secondary antibody (Figure 2E). Finally, we confirmed the specificity of the secondary antibody by exposing the tissue to this antibody only and detected no signal (Figure 2F).

### 3.3. General Observation of S1R Distribution in the Mouse Brain

Upon validation of the specificity of the primary and secondary antibodies, we executed the immunolabeling. Following IHC experiments, we first observed S1R expression in the layers of the olfactory bulb (OB), including the glomerular layer (GL), the outer plexiform layer (OPL), and the granular cell layer (GCL) (Figure 3A). S1R was also detected in the cortex (CTX), septum (SP), striatum (STR), nucleus accumbens shell and core (NAcS or NAcC), dorsal and ventral hippocampus (dHP or vHP), hypothalamus (HY), thalamus (TH), amygdala (AMG), substantia nigra (SNr), pons (P), dorsal raphe nucleus (DRN), and granular (GCL) and molecular (MCL) cell layers of the cerebellum (CB) (Figure 3B–F). Worthy of note, high signal intensity was detected in the dorsal hippocampus of juvenile male mice, especially in the CA1 region. Hence, we set this region as our internal positive control and used it as a reference for downstream analysis. Normalization to CA1 from juvenile males allows us to evaluate the relative S1R expression in a brain-region-specific manner, considering age and sex (Table 1). Assessment of S1R distribution profile in the brain within the same sex and age group showed that S1R is predominantly expressed in the hippocampus and amygdala and abundant in the shell and core region of the nucleus accumbens (Figure 4A–F). As observed, S1R is slightly detected in the hypothalamus, motor and cingulate cortex, dorsal striatum, and quite faintly expressed in the thalamus of both males and females regardless of the age group (Figure 4A–F). All female mice showed the highest expression of S1R in the CA3 region (Figure 4D–F), whereas in male mice, S1R expression in the CA3 area is greater during early adulthood only (Figure 4B). In contrast to females, S1R signal detected in the AMG was lower than in the CA1 region in juvenile and early adult male mice (Figure 4A,B). Equivalent results were observed in the AMG of middle-aged male mice when compared to female mice (Figure 4C,F).

### 3.4. S1R Expression and Distribution in the Hippocampus, Amygdala, Hypothalamus, and Thalamus

S1R expression was detected in the hippocampus, thalamus, hypothalamus, and amygdala (Figure 5A,E,G,I). As expected, S1R was detected in the perinuclear area regardless of the brain region (Figure 5B–D,F,H,J).

#### 3.4.1. The Hippocampus

The hippocampus plays a crucial role in memory and learning, serving as one of the brain’s primary centers for long-term memory storage [32]. The hippocampus includes distinct areas, including the DG, CA1, and CA3 fields (each located within the Ammon’s horns) and the subiculum. DG, CA1, and CA3 all demonstrated S1R expression, particularly in the pyramidal layer (Figure 6). As previously noticed, S1R was strongly expressed in CA1 of both male and female juvenile mice (Figure 6A,D). However, the expression decreased by nearly 50% in early adult males while remaining unchanged afterward in middle-aged mice. Instead, in female mice, the expression decreased gradually from juvenile to the middle-aged stage (Figure 6A,D–F). Similar to CA1, S1R expression in CA3 showed a gradual decrease starting in early adulthood, reaching 50% in both middle-aged males and females (Figure 6B,D–F). In the DG, S1R expression decreased by 55–65% during the transition from juvenile to early adulthood and remained stable thereafter. This pattern was consistent between male and female subjects (Figure 6C,D–F). Interestingly, our results indicate that S1R expression is not present in the subgranular layer of DG in either male or female juvenile mice but detected in early adulthood onwards (Figure 6G–J). In fact, the DG is composed of four layers: the hilum, the molecular layer (ML), the granular layer (CGL), and the subgranular zone (SGZ). Given that the latter contains interneuron cell bodies and dormant cells that undergo neurogenesis [33,34], it suggests that S1R is mainly expressed in young and mature neurons rather than quiescent and progenitor cells.

#### 3.4.2. The Amygdala

The amygdala plays a role in emotional memory, particularly in the formation and storage of fear-related memories [35], and is involved in anxiety and stress [36,37,38]. It is a brain structure composed of several nuclei, including the central (CeA), basolateral (BLA), and medial (MeA) amygdala. In our study, we focus on the BLA, which plays a leading role in the processing and regulation of emotions and sensory experiences related to fear, emotional learning, and decision-making. S1R expression was quite elevated in the amygdala of juvenile mice of both sexes (Figure 4A,D and Figure 7A,B). Although not significant, a decline of 60% in S1R expression was observed during the transition from juvenile to adulthood, which persisted throughout the aging process (Figure 7A,B).

#### 3.4.3. The Hypothalamus

The hypothalamus is a brain region described as the conductor of the endocrine and autonomic nervous systems [39]. This region is made up of several distinct nuclei, each playing a specific role. These are involved in the regulation of feeding behavior, circadian rhythm, body temperature, thirst, and control of the endocrine system, in addition to hormone release [40]. All these functions are performed by the hypothalamus in interaction with the pituitary gland, establishing vital communication between the brain and the hormonal system. As mentioned previously, S1R is slightly expressed in the hypothalamus when compared to other brain regions (Figure 4). In juvenile mice, the level of S1R expression in females exceeds by 60% the level expressed in males (Figure 7C). Again, although not significant, a decline of 50% in S1R expression was observed in male mice during the transition from juvenile to adult, which persists throughout the aging process. However, we observed a 70% decrease from juvenile to early adult in females that continued to the middle-aged stage (Figure 7C,D).

#### 3.4.4. The Thalamus

The thalamus serves as a central nervous system hub that prominently influences the integration of numerous neural activities. It receives sensory data from other nervous centers, assesses it meticulously, and consequently directs it towards the cerebral cortex. The thalamus consists principally of excitatory and inhibitory neurons. Thalamo-cortical neurons acquire motor or sensory information from diverse parts of the body and transmit it to the cerebral cortex. The thalamus is connected to the hippocampus, mammillary bodies, and fornix via the mammillo-thalamic tract [41]. A gradual age-dependent decrease in S1R expression within the thalamus was noted (Figure 7E). In fact, both males and females exhibited an estimated reduction of 40% in the early adult stage, followed by another 50% decrease from the early adult to middle-aged transition (Figure 7E,F).

### 3.5. S1R Expression and Distribution in the Cortex and Striatum

Expression of S1R was also observed in the perikaryon of the cortex (CTX), striatum, and nucleus accumbens core and shell (Figure 8 and Figure 9).

#### 3.5.1. The Motor and Cingulate Cortex

The cerebral cortex is the outer layer of the brain responsible for various cognitive and sensory processes. Comprised mainly of gray matter and structured into multiple distinct layers, it contains primarily the cell bodies of neurons. The cortex’s anatomy may be classified into various regions, each with its singular structural and functional features. Notable cortical regions are the frontal, parietal, temporal, and occipital cortex [42]. These cortical regions are connected by nerve fibers that form communication pathways, enabling the transmission of information between regions [43]. S1R is expressed in the motor and cingulate cortex (Figure 8). Similarity in S1R expression between males and females in the latter regions was also observed **(**Figure 8A,B). Regarding age impact, S1R expression levels in the motor cortex decreased gradually but sharply by 60–75% with age, remaining steady from early to advanced adulthood in both sexes (Figure 8A,C–E). A similar decreasing pattern was observed in the cingulate cortex, with a 55–80% decrease from juvenile to early and advanced adulthood (Figure 8B,C–E).

#### 3.5.2. The Striatum

The striatum consists of various nuclei in humans, including the caudate nucleus, putamen, and nucleus accumbens. In rodents, however, the striatum is subdivided into dorsal and ventral striatum. These nuclei connect with other brain regions, such as the prefrontal cortex, thalamus, internal and external globus pallidus, and substantia nigra, through complex neural circuits. The nucleus accumbens (NAc) is a structure of the basal ganglia that comprises the ventral part of the striatum. The NAc plays a crucial role in the reward system, as it is involved in motivation, reinforcement of rewarding behaviors, and the mechanisms of addiction. It is divided into two nuclei: the core (NAcC), which is associated with motor functions, and the shell (NAcS), which is associated with reward and motivation [44]. These subregions consist of 90–95% medium spiny neurons (MSNs) and 5–10% interneurons [45,46]. The core region receives projections from the motor system, while the shell receives projections from the limbic system [47]. The dorsal striatum is a structure that is also part of the basal ganglia and plays a crucial role in motor function, decision-making, and reward processing [48,49]. As observed, the expression of S1R in the NAc core and shell is similar in both males and females (Figure 9A,B). A significant reduction of 30–40% expression of S1R was detected in the NAcS in both male and female mice from the juvenile stage to early adulthood and beyond (Figure 9A,D–F). S1R expression was reduced by half in the NAcC from juvenile to middle-aged male mice, whereas no significant loss of S1R was detected in the female NAcC **(**Figure 9B,D–F). Within the dorsal striatum, the expression of S1R was again similar between sexes (Figure 9C). However, it decreased about 50% from early adulthood and beyond, indicating age-dependent changes in S1R expression (Figure 9C,G–I).

## 4. Discussion

Although there have been several reports on the effect of S1R complete deletion in the brains of male and female mice, no research has yet been devoted to its sex-related expression during normal aging. Herein, we confirm that S1R is, indeed, widely expressed throughout the brain and downregulated in several brain regions during aging in males and females. Brain macrodissection of the hippocampal, cortical, and striatal aeras followed by WB analysis of S1R levels confirmed a reduction of S1R expression within these regions for both sexes. However, sexual dimorphism was only observed in hippocampal and cortical regions, as most females had a reduced amount of S1R when compared to age-matched male mice. Our IHC results enabled us to better define the brain regions showing greater sex- and age-related differences in S1R expression. Areas such as the hippocampus and amygdala exhibited stronger S1R signals. Moderate intensities were observed in the hypothalamus, cingulate, motor cortex, and dorsal striatum, while faint intensities were observed in the thalamus (summarized in Table 1). Finally, our data indicate that, despite some relative differences, the expression pattern of S1R was mostly similar between males and females for most brain regions studied, showing stronger signal and higher variability at juvenile stages and reduced intensities through adulthood. The variability observed in S1R levels within the juvenile mice group is most likely attributed to the neuronal development and plasticity that is still occurring at this age. In fact, it is now known that brain microstructural changes in the mouse brain persist up until adulthood, i.e., 6 months of age [50]. The few significant sex differences observed in IHC analysis were noticed at the juvenile stage in the CA1 and hypothalamus. Within the CA1, juvenile females showed a reduced amount of S1R when compared to male mice, whereas in the hypothalamus, the opposite outcome was observed. Together, these observations suggest a potential sexual dimorphic pattern of S1R distribution before puberty in the hypothalamus and the hippocampus, which was partially supported by the WB experiments. However, in-depth research is necessary to have a complete understanding of this observation.

Numerous studies have described S1R expression in different regions of the brain with changing intensities. Several techniques have been used in different animal models for these purposes. Currently, data on age-related changes in S1R expression are inconclusive. Nevertheless, the detection of S1R in monkeys using [^11^C]SA4503 has been made possible by PET scans. Based on the kinetic parameters of this tracer, the highest binding potential was found in the hippocampus [25]. This aligns with our findings showing that S1R exhibits high expression specifically in the pyramidal layer of the CA1 and CA3 fields regardless of age. Among young monkeys, binding potential values were moderate in the frontal cortex, thalamus, cerebellar hemisphere, striatum, temporal cortex, vermis, and parietal cortex and were comparatively low in the occipital cortex. Age-related discrepancies in binding potential were significant in older monkeys. In fact, S1R expression is enhanced in older monkeys in comparison to young primates [25]. Another study examined age-related changes in S1R expression in rats through PET scans, again with the S1R-specific radioligand [^11^C]SA4503 [1]. The results showed that radioactivity levels were higher in the pons, medulla, and spinal cord, followed by the midbrain, thalamus, and hypothalamus in juvenile rats. Lower levels of radioactivity were observed in several brain regions, including the cortex, striatum, hippocampus, olfactory bulbs, amygdala, and cerebellum. According to Ramakrishnan et al., the impact of age on the radioligand uptake varies across different regions in rats, with some areas exhibiting greater changes than others [1]. This diverges from our results, which suggest a consistent decrease in S1R with age across all brain regions. However, one must keep in mind that the radioligand used in these studies will associate and detect S1R only if the binding pocket is accessible [51]. This contrasts with an antibody-based detection method that can recognize and detect a ligand-bound protein. Given that (1) neurosteroids are among the most important physiological ligands of S1R [51], that (2) steroids binding domain loop (SBDL) I and II are comprised in the S1R binding pocket [52,53,54], and that (3) neurosteroid biogenesis is reduced in the aging brain [55,56], the enhanced detection of S1R might be explained by an increased availability of the binding pocket, which is most likely free of its endogenous ligand. Additionally, cerebral blood flow plays a crucial role in the distribution of the labeled radiotracer. Age-related changes in the brain can affect both metabolic activity and blood flow [57]. These changes, along with the detection system used, may explain the differences in signal intensity observed on PET scans compared to our findings.

DAB-Immunohistochemistry staining performed in mice aged 2 months (young adults) and 24 months (aged) revealed a substantial amount of S1R detected in the olfactory bulbs, cerebral cortex, hippocampus, hypothalamus, and cerebellum, with subsequent accumulation in the striatum, septum, nucleus accumbens, and amygdala [26]. In the present study, we found that S1R expression was perinuclear in all brain regions examined, which is consistent with findings reporting perikaryon expression of S1R in the olfactory bulbs, granular layer, and cells of the glomerular layer [26]. Furthermore, S1R expression is present in the granular layer of the cerebellum but absent in the Purkinje layer. This same report concluded that the level of S1R expression remains stable across various brain regions with age [26]. Our study shows that S1R is significantly expressed in the hippocampus, particularly in neurons located in the pyramidal layer of CA1 and CA3 subfields, and in the granular layer of DG. Phan and colleagues also observed a high intensity expression of S1R in the hippocampus [26]. The signal intensity was detected in neurons located in the pyramidal layer of the CA1 and CA3 regions and the granular layer of the DG [26]. Our findings demonstrate that S1R expression decreases abruptly after the juvenile stage in all subregions, i.e., CA1, CA3, and DG, in both male and female mice and both techniques (WB and IHC). One could speculate that the age-related reduction of S1R expression in the hippocampus might promote memory deficits. In fact, Phan et al. have shown that agonist-induced activity of S1R enhances Morris-water maze performance of aged mice [26], while Crouzier et al. showed that inhibition of S1R activity or deletion of S1R in 2-month-old mice decreases complex-cognitive task performance [58]. Additionally, we did not witness any significant variation between early adult and middle-aged mice, which is consistent with the data reported by Phan et al. Interestingly, we noticed that S1R was not expressed in the DG’s subgranular zone of juvenile mice from both sexes. However, S1R distribution in this zone was enhanced during aging. It is known that the subgranular zone contains a substantial number of quiescent cells, which play a role in neurogenesis [59]. Quiescent progenitor cells have a temporary state marked by a considerable reduction in their metabolic activities. This decrease in cellular activity leads to reduced energy consumption and metabolic waste production. Research has demonstrated that the quantity of quiescent cells in the subgranular zone diminishes exponentially as one ages [60]. As individuals grow older, quiescent neural progenitor tends to decrease in the hippocampus. This decline is a typical and innate phenomenon related to aging [61]. Our data indicate that the lack of S1R expression in the subgranular zone in juvenile males and females could be attributed to the prevalence of quiescent cells in the latter. The reduced activity of these quiescent cells and their low metabolic stress level is thus consistent with the lack of S1R expression in these cells.

The mechanisms responsible for the reduced expression of S1R during brain aging have yet to be fully characterized, but several explanations may account for it. Aging is generally associated with a decrease in gene expression across all tissues, including the brain [57]. For instance, research has shown a decrease in the expression of specific neurotransmitter receptors, including NMDA receptors, in rats, mice, and monkeys [62,63]. In addition, the efficiency of the set of cellular mechanisms responsible for maintaining the proteome homeostasis decreases with age [64]. The maintenance of proteostasis and the prevention of protein aggregate formation in the ER are ensured by a complex network, including the unfolded protein response (UPR) pathway, which involves chaperone proteins and proteolytic machinery. The UPR pathway includes effector proteins like PERK (PKR-like ER kinase), ATF6 (activating transcription factor 6), and IRE1; the latter is stabilized by S1R during ER stress [65]. The involvement of S1R in these signaling pathways, which become less effective with age, could account for the reduction in S1R expression that occurs during aging. Overall, our findings provide valuable insights into the spatiotemporal dynamics of S1R expression in the brain, highlighting its potential relevance in the context of developmental and age-related processes in both sexes, as well as its implications for neurological function and pathology.

## Figures and Tables

**Figure 1 brainsci-14-00881-f001:**
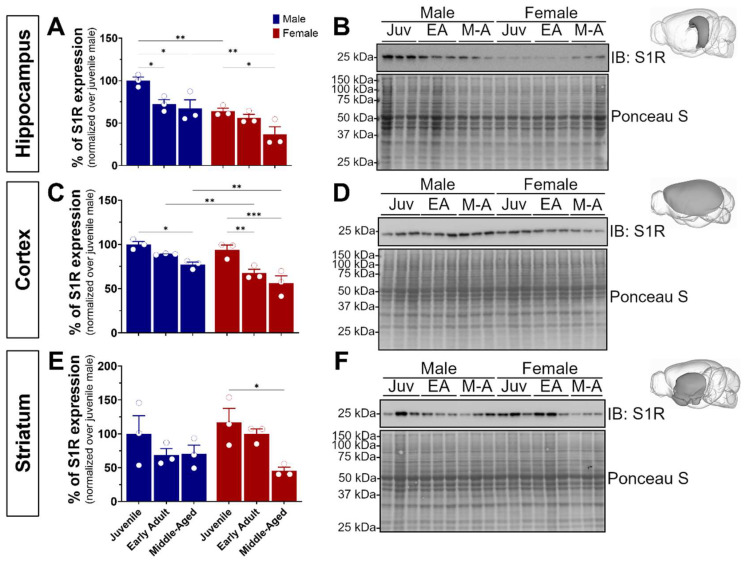
Age-dependent decline of S1R protein level was observed in hippocampal, cortical, and striatal tissue. Expression of S1R in the hippocampus (**A**,**B**), isocortex (**C**,**D**), and striatum (**E**,**F**) of juvenile, early adult, and middle-aged male and female mice. The blots shown are representative of three separate experiments. The 3D brain insets were from the Scalable Brain Atlas [31]. The S1R antibody used is from CST cat#61994S, 1/500. The statistical significance was determined using two-way ANOVA, followed by a post hoc Bonferroni multiple comparison test. Data were considered significant when *p* values were <0.05 (*), <0.01 (**), or <0.001 (***). See Appendix A for original blots.

**Figure 2 brainsci-14-00881-f002:**
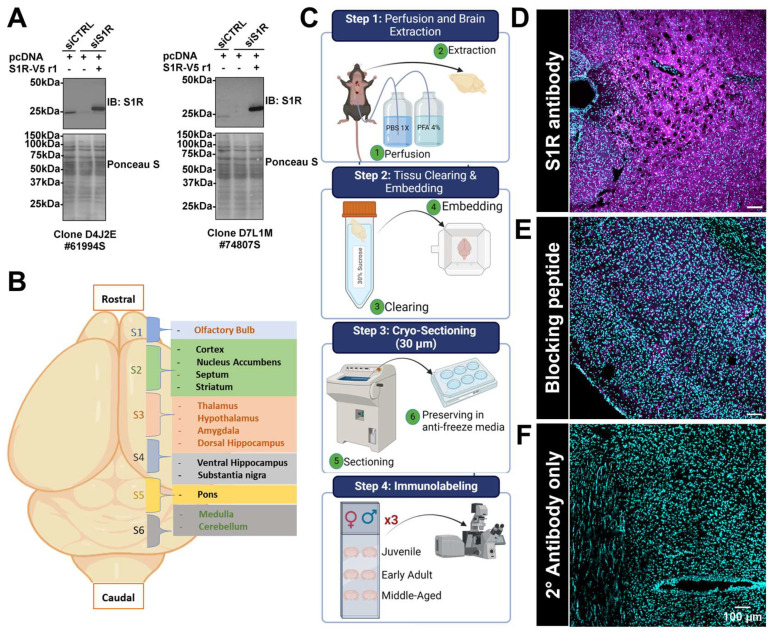
Validation of S1R primary antibodies’ specificity and synoptic diagram of the IHC methodology and specificity of the primary and secondary used for IHC. In vitro validation of S1R primary antibodies specificity. The blots shown are representative of two separate experiments. See Appendix A for original blots (**A**). Identification of the different sections and brain regions analyzed in this study (**B**). The representative scheme of the protocol used to perfuse, extract, section, embed, mount, label, and visualize the mouse brain (**C**). Labeling of S1R in juvenile male brain tissue, using the primary antibody rabbit anti-S1R (#74807S) and goat anti-rabbit Alexa Fluor^®^ 647 conjugate as the secondary antibody (**D**). Prior to labeling, the S1R primary antibody was incubated overnight at 4 °C with a blocking peptide, i.e., a recombinant protein fragment of S1R-Ctail, with end over end mixing. Immunolabeling was then performed as mentioned in the methods (**E**). Brain sections were only exposed to the secondary antibody (**F**). S1R is shown in magenta, while nuclei are in cyan.

**Figure 3 brainsci-14-00881-f003:**
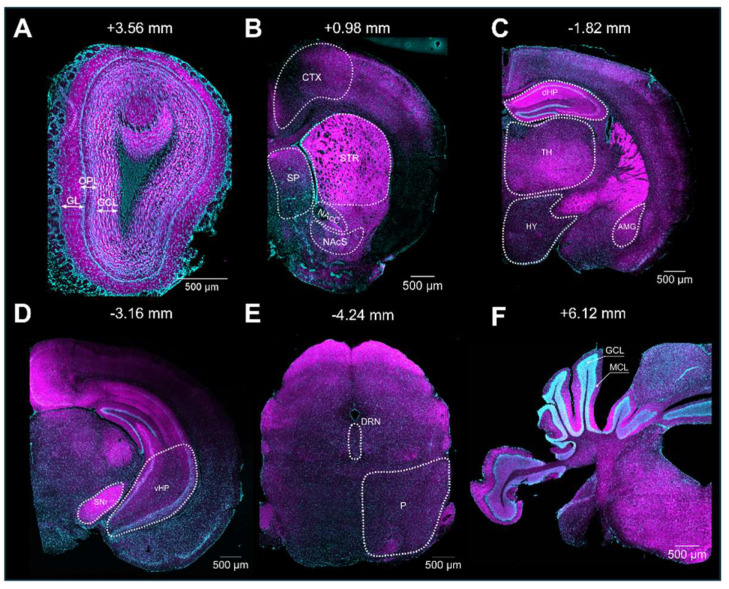
Expression of S1R in the mouse brain. Rostro-caudal coronal sections of a juvenile male mouse brain. The S1R signal is shown in magenta, while the nuclei are in cyan. The signal was observed across diverse layers of the olfactory bulb (OB), including the glomerular, outer plexiform, and granular layers (abbreviated as GL, OPL, and GCL, respectively) (**A**). CTX, Cortex; SP, septum; STR, striatum; NAcS, nucleus accumbens shell; NAcC, nucleus accumbens core (**B**). dHP, dorsal hippocampus; HY, hypothalamus; TH, thalamus; and AMG, amygdala (**C**). SNr, substantia nigra; and vHP, ventral hippocampus (**D**). P, pons; and DRN, dorsal raphe nucleus (**E**). CB; cerebellum; GCL, granular layer; and MCL, molecular layer (**F**).

**Figure 4 brainsci-14-00881-f004:**
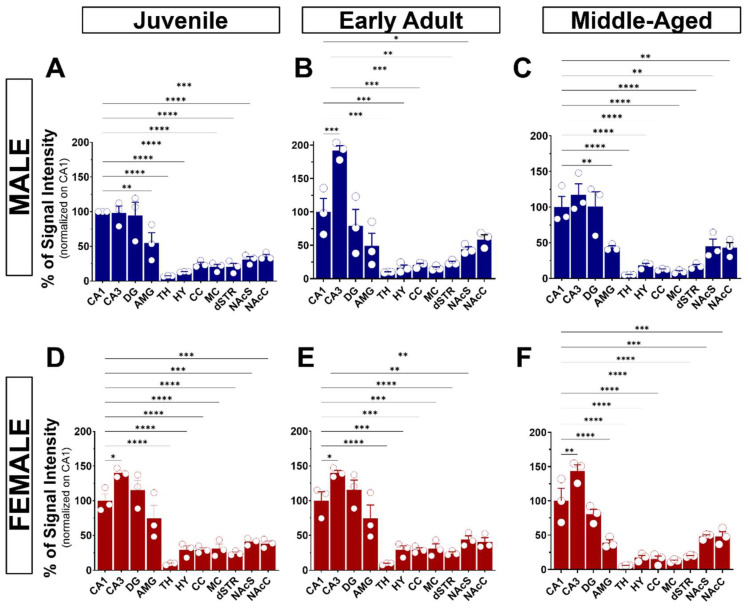
Expression of S1R in different brain regions of age-matched male and female mice. The data were normalized on the CA1 signal intensity and represent the mean ± SEM values (n = 3 biological replicates each analyzed in 3 technical replicates). Statistical significance was evaluated using a one-way ANOVA followed by Bonferroni post hoc multiple comparisons test; * *p* ≤ 0.05, ** *p* ≤ 0.01, *** *p* ≤ 0.001, **** *p* ≤ 0.0001. CA1 and CA3, Cornu Ammonis; DG, dentate gyrus; AMG, amygdala; TH, thalamus; HY, hypothalamus; CC, cingulate cortex; MC, motor cortex; dSTR, dorsal striatum; NAcS, nucleus accumbens shell; NAcC, nucleus accumbens core.

**Figure 5 brainsci-14-00881-f005:**
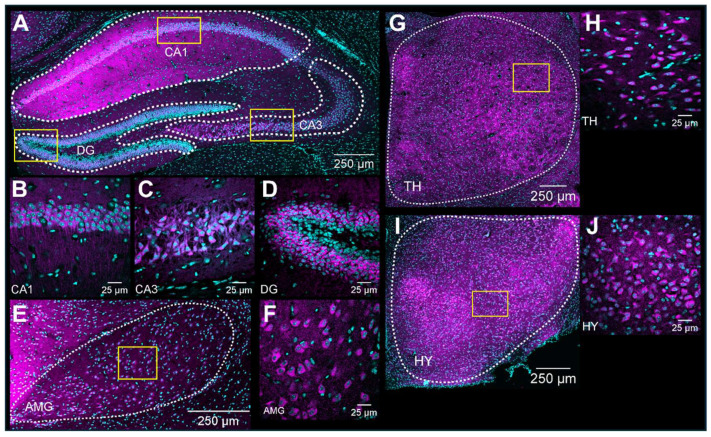
Expression profile of S1R in the hippocampus, amygdala, hypothalamus, and thalamus. Expression of S1R in dHP subregions, including CA1, CA3, and DG (**A**). Representative confocal images displaying perinuclear localization of S1R in the hypothalamus, thalamus, amygdala, and hippocampal subregions CA1, CA3, and DG. (**B**–**J**). S1R is shown in magenta, while nuclei are in cyan.

**Figure 6 brainsci-14-00881-f006:**
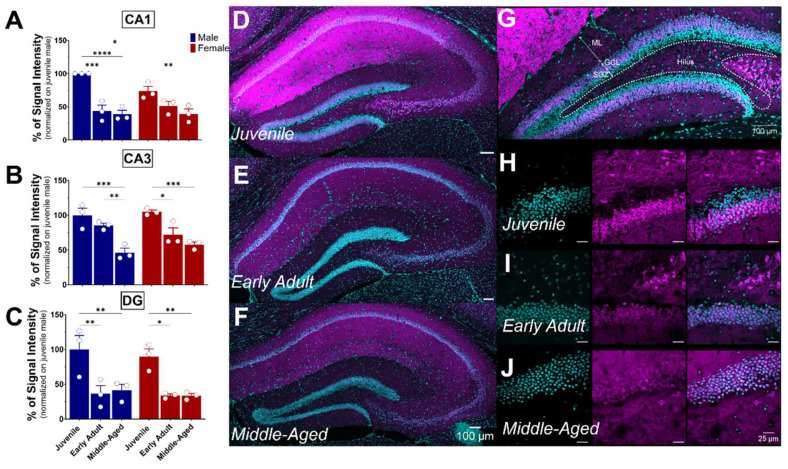
Ontogeny of S1R expression in the hippocampus CA1, CA3, and DG subregions. Signal intensity of S1R in CA1, CA3, and DG (n = 3 slices/section/brain, n = 3 mice per group). The data present the mean ± SEM values. Statistical significance was evaluated using a two-way ANOVA followed by Bonferroni post hoc multiple comparisons test; * *p* ≤ 0.05, ** *p* ≤ 0.01, *** *p* ≤ 0.001, **** *p* ≤ 0.0001 (**A**–**C**). Representative images displaying S1R expression in the dorsal hippocampus of juvenile, early adult, and middle-aged mice. (**D**–**F**). Representative images of a juvenile male mouse DG and identification of the DG’s subregions. ML, molecular layer; GCL, granular cell layer; SGZ, subgranular zone (**G**). Representative images of S1R localization in SGZ and GCL at different ontogenic stages. (**H**–**J**). S1R is shown in magenta, while nuclei are in cyan.

**Figure 7 brainsci-14-00881-f007:**
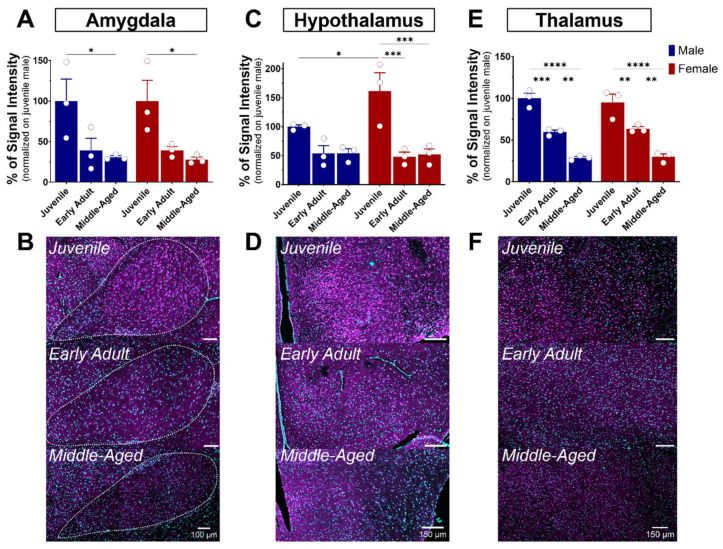
Ontogeny of S1R level in the amygdala, hypothalamus, and thalamus in male and female mice. Signal intensity of S1R in AMG, HY, and TH (n = 3 slices/section/brain, n = 3 mice per group). The data represent the mean ± SEM values. Statistical significance was evaluated using a two-way ANOVA followed by Bonferroni post hoc multiple comparisons test; * *p* ≤ 0.05, ** *p* ≤ 0.01, *** *p* ≤ 0.001, **** *p* ≤ 0.0001 (**A**,**C**,**E**). Representative pictures of coronal brain sections of juvenile, early adult, and middle-aged mouse AMG (**B**). HY (**D**). and TH (**F**). S1R is shown in magenta, while nuclei are in cyan.

**Figure 8 brainsci-14-00881-f008:**
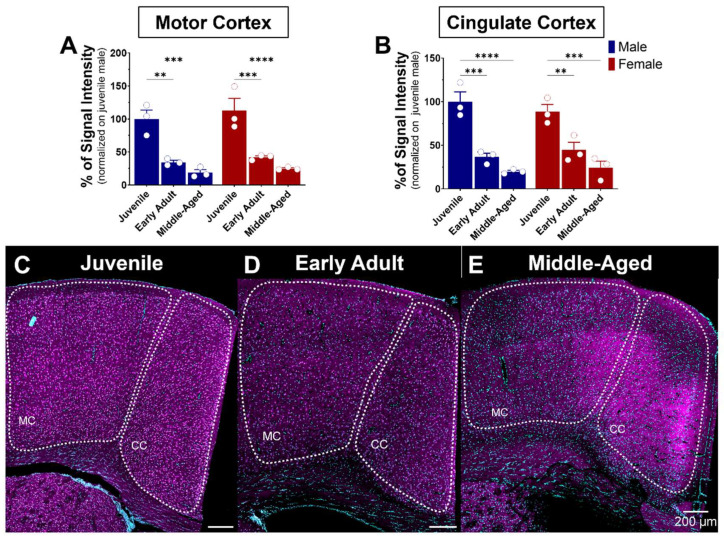
Age-related expression of S1R in the motor and cingulate cortex of male and female mice. Signal intensity of S1R in MC and CC (n = 3 slices/section/brain, n = 3 mice per group). The data represent the mean ± SEM values. Statistical significance was evaluated using a two-way ANOVA followed by Bonferroni post hoc multiple comparisons test; ** *p* ≤ 0.01, *** *p* ≤ 0.001, **** *p* ≤ 0.0001 (**A**,**B**) Representative images displaying S1R in the MC and CC of juvenile, early adult, and middle-aged mice (**C**–**E**). S1R is shown in magenta, while nuclei are in cyan.

**Figure 9 brainsci-14-00881-f009:**
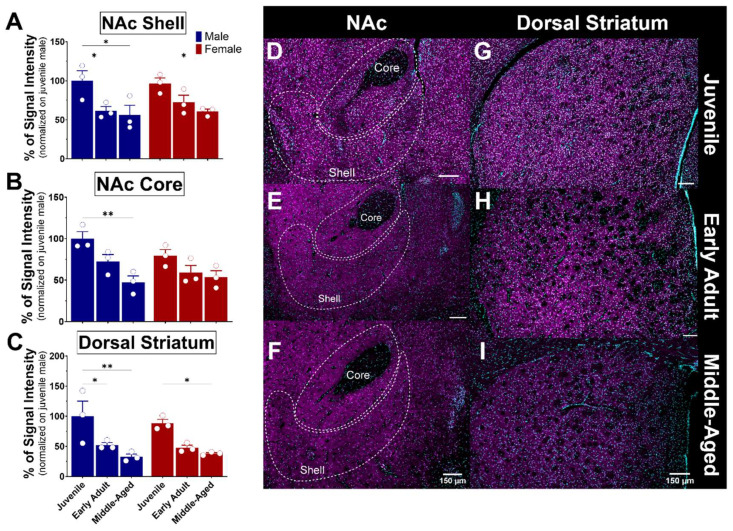
Ontogeny of S1R expression in the nucleus accumbens and dorsal striatum of male and female mice. Signal intensity of S1R in the NAcS, NAcC, and dSTR (n = 3 slices/section/brain, n = 3 mice per group). The data represent the mean ± SEM values. Statistical significance was evaluated using a two-way ANOVA followed by Bonferroni post hoc multiple comparisons test; * < *p* ≤ 0.05, ** *p* ≤ 0.01. (**A**–**C**) Representative images displaying S1R in the NAc and dSTR of juvenile, early adult, and middle-aged mice (**D**–**I**). S1R is shown in magenta, while nuclei are in cyan.

**Table 1 brainsci-14-00881-t001:** Normalized S1R signal intensity detected across brain regions.

Brain Regions	Female			Male		
	Juvenile	Early Adult	Middle-Aged	Juvenile	Early Adult	Middle-Aged
Hippocampus						
CA1	73.56 ± 7.13	51.59 ± 6.61	39.61 ± 7.25	100	43.72 ± 8.75	38.94 ± 5.87
CA3	103.07 ± 2.74	70.91 ± 9.5	56.87 ± 3.58	98.19 ± 10.02	83.83 ± 3.26	45.64 ± 6.03
DG	84.91 ± 10.74	32.09 ± 2.10	31.83 ± 2.93	94.54 ± 18.97	34.55 ± 10.75	39.27 ± 8.02
Thalamus	6.86 ±0.73	4.58 ± 0.19	2.15 ± 0.25	7.22 ± 0.44	4.30 ± 0.16	2.06 ± 0.11
Amygdala	55.02 ± 13.97	21.60 ± 2.56	15.34 ± 1.76	54.95 ± 14.87	21.46 ± 8.29	16.83 ± 1.06
Hypothalamus	21.52 ± 4.20	6.42 ± 1.05	6.92 ± 1.28	13.33 ± 0.41	7.15 ± 1.81	7.17 ± 1.06
Cortex						
Cingulate	21.61 ± 2.05	10.94 ± 2.09	5.94 ± 1.84	24.42 ± 2.74	8.96 ± 1.04	4.87 ± 0.35
Motor	22.95 ± 4.84	8.40 ± 0.6	4.98 ± 0.49	20.42 ± 3.83	6.95 ± 0.81	3.66 ± 0.57
Striatum						
Dorsal	18.01 ± 1.33	9.74 ± 0.86	7.78 ± 0.31	20.38 ± 5.12	10.65 ± 0.84	6.69 ± 0.90
NAc Shell	30.22 ± 2.20	22.66 ± 2.83	18.98 ± 1.00	31.27 ± 4.05	19.22 ± 1.76	17.59 ± 3.89
NAc Core	28.21 ± 2.65	20.94 ± 3.11	19.05 ± 2.75	35.55 ± 2.98	25.75 ± 2.97	16.80 ± 2.78

## Data Availability

The raw data supporting the conclusions of this article will be made available by the authors on request.

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
