# Peer review of "Comprehensive Analysis of Age- and Sex-Related Expression of the Chaperone Protein Sigma-1R in the Mouse Brain"

_brainsci, 2024, doi:10.3390/brainsci14090881_

Round 1

Reviewer 1 Report

Comments and Suggestions for Authors

The author has made a commendable effort to illustrate the distribution of Sigma-1R expression in the mouse brain; however, several aspects require clarification to substantiate the hypothesis.

·         Notably, the spelling of "several" in the abstract (line #16) needs correction, and reference #1 (Bakker et al., 2015) appears to be missing.

·         In the results section (#3.1), the bar graph depicting hippocampal data for females indicates minimal differences between juvenile and early adult groups, raising questions about the significance of these findings. Furthermore, the striatum graph for males shows considerable variability among juvenile biological replicates, complicating the assertion of significance when compared to early adult and middle-aged groups.

·         The author should also consider presenting gene expression data for Sigma-1R.

·         In section #3.2, clarification is needed regarding the age of mice used for histological staining.

·         Additionally, the author could explore the relationship between Sigma-1R expression and memory regulation, particularly in the context of aging, as memory decline may correlate with reduced Sigma-1R levels.

·         Expanding the study to include the effects of blocking Sigma-1R on male and female brain development related to memory cell growth could provide valuable insights.

·         Overall, the results suggest minimal differences in Sigma-1R expression between males and females, except in the hypothalamus and NAC core, where all most same expression pattern is observed across regions.

Author Response

Reviewer #1

The author has made a commendable effort to illustrate the distribution of Sigma-1R expression in the mouse brain; however, several aspects require clarification to substantiate the hypothesis.

  1. Notably, the spelling of "several" in the abstract (line #16) needs correction, and reference #1 (Bakker et al., 2015) appears to be missing.

We thank the reviewer for noticing the mistake and confirm that the spelling of the word several on line #16 was corrected. The reference #1 (Bakker et al. 2015) is cited in the figure legend 1, line #233.

  1. In the results section (#3.1), the bar graph depicting hippocampal data for females indicates minimal differences between juvenile and early adult groups, raising questions about the significance of these findings.

Indeed, the bar graph depicting hippocampal data for female in figure 1, section #3.1, does not show any significant statistical difference between the juvenile and early adult group as opposed to IHC data shown in figure 6, section #3.4.1.  As mentioned in the text (line #219-220), we harvested and extracted the protein from the entire hippocampus and have not microdissected the CA1, CA3 or DG region. While this approach inherently yields to greater variability, it provided us with preliminary evidence of reduced S1R expression in this brain area. Therefore, to achieve higher accuracy, spatial resolution, an reduced variability, we chose to carry out IHC experiments. This approach allows us to precisely delineate hippocampal areas, i.e. CA1, CA3 and DG, and quantify S1R expression in these subregions (as mentioned in the text, line #238-239). Importantly, IHC allowed us to identify specific change in the granular cell layer of the DG (Fig. 6G-J), reaching a spatial resolution that is not possible with WB analyses. Taken together, these data support that S1R expression is significantly reduced when female mice transition from juvenile to early adult age. We hope that these justifications meet with the reviewer’s approval.

Furthermore, it is important to note that WB experiments ensure that the observed changes in immunofluorescence were not caused by perfusion artifacts or differences in antibody tissue penetration across ages, thereby enhancing the rigor and reliability of our study.

Furthermore, the striatum graph for males shows considerable variability among juvenile biological replicates, complicating the assertion of significance when compared to early adult and middle-aged groups.

Indeed, high variability is observed amongst juvenile male biological replicates. As mentioned in the text (line #225-227), although not significant, we detected an age-related decline of S1R expression in male striatal tissue. As for the previous comment, the WB analysis provided us with preliminary evidence of reduced S1R expression in the striatum, and we decided to carry out IHC experiments to better delineate the striatal areas, i.e. dorsal striatum, NAc core and NAc shell, and quantify S1R expression in these subregions (as mentioned in the text, line #238-239). Overall, despite the observed high variability, the data shown in the current study support that S1R expression significantly declines in the striatum of male mice with aging.

  1. The author should also consider presenting gene expression data for Sigma-1R.

We thank the reviewer for the suggestion and acknowledge the value of mapping Sigmar1 gene expression in the brain. We hope that the reviewer understands that performing ISH experiments using these conditions (sex and age) will require a tremendous amount of time and resources, which are beyond the scope of the current study. Although incorporating RNA data could be interesting, it is unlikely to substantially improve the impact of the findings reported in this study.

  1. In section #3.2, clarification is needed regarding the age of mice used for histological staining.

We thank the reviewer for noticing the missing information and confirm that we have now included this info if the legend of figure 2 (see line # 267).

  1. Additionally, the author could explore the relationship between Sigma-1R expression and memory regulation, particularly in the context of aging, as memory decline may correlate with reduced Sigma-1R levels.

We thank the reviewer for the suggestion and understand that the relationship between S1R expression and memory regulation is of interest, especially in the aging context. We hope the reviewer understands that executing this experiment would require several years and substantially more resources, making it beyond the scope of the current study.

  1. Expanding the study to include the effects of blocking Sigma-1R on male and female brain development related to memory cell growth could provide valuable insights.

As for the previous comment, we thank the reviewer for the suggestion and understand that the relationship between S1R activity level in the developing brain of male and female and its impact on cognition is of interest. However, conducting such experiments would require extensive time and resources, well beyond the scope of this current study. We hope the reviewer appreciates these constraints.

  1. Overall, the results suggest minimal differences in Sigma-1R expression between males and females, except in the hypothalamus and NAC core, where all most same expression pattern is observed across regions.

Indeed, we agree with the reviewer’s comment. As we mentioned in the text, line #467-470, section #4, our data indicate that, despite some relative differences, the expression pattern of S1R was mostly similar between males and females for most brain regions studied, showing stronger signal at juvenile stages and a reduced intensities through adulthood.

Reviewer 2 Report

Comments and Suggestions for Authors

The overall experimental design is in good logic but with some consideration on sample size. In summary, my suggestion would be reconsider after a major revision. The questions and detailed comments are as follows:

1. Line 63~64, the summary of project finding: Is the conclusion of this manuscript there is "sex-specific impact on the expression of S1R in the aging mouse brain" ? If it so, is there any evidence of previous work show that sex is an important factor for studying mouse brain or relative disorders? This is kind of unclear in the introduction part. Same as the summary of the whole work.

2. The sample size of each age stage is 3 male and 3 female. According to different results across the whole study, there are obvious individual bias and long error bars for both male and female in juvenile mice through R1 expression to signal intensity. It would not convincible to me when there are only 3 samples that with such obvious unstable results within a group. Can you please explain on this with more details? Or maybe add more samples for the experiment.

3. Please double check on some minor error and typo. e.g line 16: "sev-eral" -> "several"

Author Response

Reviewer #2

The overall experimental design is in good logic but with some consideration on sample size. In summary, my suggestion would be reconsidered after a major revision. The questions and detailed comments are as follows:

  1. Line 63~64, the summary of project finding: Is the conclusion of this manuscript there is "sex-specific impact on the expression of S1R in the aging mouse brain»? If it so, is there any evidence of previous work show that sex is an important factor for studying mouse brain or relative disorders? This is kind of unclear in the introduction part. Same as the summary of the whole work.

We thank the reviewer for the comment. We changed the sentence used in line #63-64, as it was meant to define the aim of our study and not its conclusions. Thus, we modify the previous sentence for the following (line #62-65): “Considering the limited research on sex-specific variability in S1R expression and the inconclusive findings regarding age-related changes in S1R levels in the brain, our study seeks to determine the age- and sex-specific effects on S1R expression in the mouse brain”.

We also refine the summary of our work in the abstract and change the previous sentence “We observed that dynamic changes are evident in juvenile mice, and most brain regions show an abrupt or gradual decline in S1R level as mice transition to adulthood” for the following: “We observed dynamic changes in S1R levels during development, with most brain regions showing either an abrupt or gradual decline as mice transition from juveniles to adults” (line# 25-27). We now hope that these changes meet with the reviewer’s approval.

  1. The sample size of each age stage is 3 male and 3 female. According to different results across the whole study, there are obvious individual bias and long error bars for both male and female in juvenile mice through R1 expression to signal intensity. It would not be convincible to me when there are only 3 samples that with such obvious unstable results within a group. Can you please explain this in more detail? Or maybe add more samples for the experiment.

We appreciate the reviewer’s suggestion regarding increasing the sample size. While we agree that expanding the number of biological replicates could be beneficial, doing so rigorously across all groups would require substantial additional time and resources. As mentioned in the methods, line #178-180, all three biological replicates were analyzed in three technical replicates for IHC analysis. Given that our current data already demonstrate clear and statistically significant differences, we believe that further replication would be unlikely to enhance or change the overall conclusions of the study.

  1. Please double check for some minor errors and typo. e.g line 16: "sev-eral" -> "several"

We thank the reviewer for noticing the mistake and confirm that the spelling of the word several on line #16 was corrected.

Reviewer 3 Report

Comments and Suggestions for Authors

Reviewer:

This study reported the S1R expression profile of age- and sex-related effects in mouse brains during developmental stages through immunostaining. The author finds that S1R is expressed in juvenile mice hippocampus, especially in CA1 and CA3 regions. In addition, S1R is not observed in the sub-granular layer of the dentate gyrus of juvenile mice. While the data are overall presented, the paper written is very well, there are several critical points for the author’s conclusion, which need the author’s attention. 

1. How do I know the S1R antibody is specifically recognized S1R? The author needs to test the antibody specifically recognized S1R in vitro or in vivo. 

2. If the author could determine the difference in the expression of S1R exhibited in excitatory neurons, interneurons, microglia, or astrocytes in terms of age- and sex-related effects, that would be more interesting!

3. Line 16, deleted the dish "-".

4. All the labeling should be consistent! Such as lines 351-352 and line 406. In addition, some used the capital with bold, but some did not. All should be consistent. 

Author Response

analysis. Given that our current data already demonstrate clear and statistically significant differences, we believe that further replication would be unlikely to enhance or change the overall conclusions of the study.

  1. Please double check for some minor errors and typo. e.g line 16: "sev-eral" -> "several"

We thank the reviewer for noticing the mistake and confirm that the spelling of the word several on line #16 was corrected.

Reviewer #3

This study reported the S1R expression profile of age- and sex-related effects in mouse brains during developmental stages through immunostaining. The author finds that S1R is expressed in juvenile mice hippocampus, especially in CA1 and CA3 regions. In addition, S1R is not observed in the sub-granular layer of the dentate gyrus of juvenile mice. While the data are overall presented, the paper written is very well, there are several critical points for the author’s conclusion, which need the author’s attention. 

  1. How do I know if the S1R antibody is specifically recognized as S1R? The author needs to test the antibody specifically recognized S1R in vitro or in vivo. 

We thank the reviewer’s comment and acknowledge the importance of validating the specificity of the antibody used in this study. We have now included siRNA assay combined with western blotting analysis to demonstrate that the S1R antibodies used are able to recognize the endogenous protein and transfected S1R cDNA plasmid. This data set is now shown in Figure 2A. Given the addition of this result, we now include Ms. Sara-Maude Bélanger et M. Guillaume Beaucaire as author of the manuscript since they both performed these experiments.

  1. If the author could determine the difference in the expression of S1R exhibited in excitatory neurons, interneurons, microglia, or astrocytes in terms of age- and sex-related effects, which would be more interesting!

We appreciate the reviewer’s suggestion and recognize the importance of studying S1R expression in these cell types with respect to age- and sex-related effects. However, conducting such experiments would require considerable time and resources, which are beyond the scope of this study. We hope the reviewer understands these limitations.

  1. Line 16 deleted the dish "-".

We thank the reviewer for noticing the mistake and confirm that the dash sign was removed and the spelling of the word several on line #16 corrected.

  1. All the labeling should be consistent! Such as lines 351-352 and line 406. In addition, some used the capital with bold, but some did not. All should be consistent. 

We appreciate the reviewer for noting these discrepancies and confirm that we have standardized the figure legend labeling accordingly.

Round 2

Reviewer 1 Report

Comments and Suggestions for Authors

The author has addressed most of the feedback from the first review, which is commendable. However, there are still a few points that need attention. For instance, the author acknowledges in the conclusion that there is no significant difference between males and females, yet the title of the article suggests a sex-specific focus. Given that the observed differences are more related to age and brain region rather than sex, the title may not accurately reflect the content.

Additionally, as previously mentioned, there might be a connection between memory cells and SR1 expression, particularly since SR1 expression decreases with age in a manner similar to memory cell changes. The author should consider discussing this potential link in the article.

Round 2

The author has addressed most of the feedback from the first review, which is commendable. However, there are still a few points that need attention. For instance, the author acknowledges in the conclusion that there is no significant difference between males and females, yet the title of the article suggests a sex-specific focus. Given that the observed differences are more related to age and brain region rather than sex, the title may not accurately reflect the content.

We thank the reviewer for the suggestion; however, we would like to clarify that the current title does not suggest that sex differences were found, but rather that both age- and sex-related expressions were analyzed. We believe that the absence of significant sex differences should not justify removing "sex" from the title. Doing so could misrepresent the comprehensive scope of our analysis. Additionally, retaining "sex" in the title is important for ensuring accurate searchability and visibility in scientific databases, particularly for those specifically seeking sex-related studies.

Additionally, as previously mentioned, there might be a connection between memory cells and SR1 expression, particularly since SR1 expression decreases with age in a manner similar to memory cell changes. The author should consider discussing this potential link in the article.

We appreciate the reviewer’s suggestion and have now discussed the correlation between the age-dependent reduction of S1R expression and cognitive impairment in the discussion’s section, lines # 615-619.

Reviewer 2 Report

Comments and Suggestions for Authors

The question 2 was based on the obvious bias between one sample from two others. A more detailed discussion or explain may need to be added.

"The sample size of each age stage is 3 male and 3 female. According to different results across the whole study, there are obvious individual bias and long error bars for both male and female in juvenile mice through R1 expression to signal intensity. It would not be convincible to me when there are only 3 samples that with such obvious unstable results within a group. Can you please explain this in more detail? Or maybe add more samples for the experiment."

Round 2

Question 2 was based on the obvious bias between one sample from two others. A more detailed discussion or explain may need to be added.

"The sample size of each age stage is 3 male and 3 female. According to different results across the whole study, there are obvious individual bias and long error bars for both male and female in juvenile mice through R1 expression to signal intensity. It would not be convincible to me when there are only 3 samples that with such obvious unstable results within a group. Can you please explain this in more detail? Or maybe add more samples for the experiment."

As mentioned previously, we appreciate the reviewer’s suggestion regarding increasing the sample size. While we agree that expanding the number of biological replicates could be beneficial, doing so rigorously across all groups would require substantial additional time and resources. As mentioned in the methods, lines #221-223, all three biological replicates were analyzed in three technical replicates for IHC analysis. Given that our current data already demonstrate clear and statistically significant differences, we believe that further replication would be unlikely to enhance or change the overall conclusions of the study. However, we now included an explanation regarding the variability observed in the discussion section, lines #551-556.

Reviewer 3 Report

Comments and Suggestions for Authors

The author addressed all my concerns!

Round 2

The author addressed all my concerns!

We thank the reviewer for the time spent on the revision of our manuscript.